# I’m More Prosocial than Others: Narcissism Facilitates Prosocial Behavior in Public Situations

**DOI:** 10.3390/bs14121200

**Published:** 2024-12-14

**Authors:** Yining Song, Qiong Chen, Ping Ren, Jing Ma, Caina Li

**Affiliations:** 1School of Psychology, Shaanxi Normal University, Xi’an 710062, China; songyining@snnu.edu.cn (Y.S.); qiongchen@snnu.edu.cn (Q.C.); talama135@snnu.edu.cn (J.M.); 2Collaborative Innovation Center of Assessment toward Basic Education Quality, Beijing Normal University, Beijing 100875, China; renping@bnu.edu.cn

**Keywords:** trait/state narcissism, prosocial behavior, public/anonymous situation, self-enhancement

## Abstract

Although narcissism consistently predicts maladaptive behaviors, the relationship between narcissism and prosocial behavior remains controversial due to the paradoxical nature of narcissism. In three studies, we investigated the roles of situational and motivational factors in the link between narcissism and prosocial behavior. In Study 1 (*n* = 932), a questionnaire survey revealed that trait narcissism positively predicted prosocial behavior tendencies only in public situations as opposed to anonymous situations. In Study 2 (*n* = 200) and 3 (*n* = 170), we manipulated individuals’ state narcissism through story-based primes and measured prosocial behavior via actual monetary donations. Both Studies 2 and 3 consistently demonstrated that participants in the state narcissism group exhibited greater prosocial behavior in public situations compared to anonymous ones. Furthermore, the findings of Study 3 indicated that the increased prosocial behavior observed in public situations could be attributed to heightened self-enhancement. These findings contribute to a more nuanced understanding of the relationship between narcissism and prosocial behavior.

## 1. Introduction

The proliferation of social media has prompted significant reflection on the intricate relationship between narcissism and prosocial behavior, as charitable behaviors are posted in high profile on social media not only by celebrities but also by general internet users. These visible acts often raise questions regarding their true motivations: are they genuinely driven by altruism, or do they primarily serve to fulfill their own narcissistic desires? For instance, research on the “Ice Bucket Challenge” reveals that participants who shared videos of their involvement exhibited heightened levels of narcissism [1]. This finding suggests that some prosocial behaviors may, in fact, represent a form of “self-display” [2], closely associated with narcissistic characteristics [3].

Narcissism—a term derived from the Greek mythological figure Narcissus, who fell in love with his own reflection [4]—involves a sense of arrogance and grandiosity, an inflated feeling of entitlement and superiority, as well as an excessive need for admiration [5]. Traditionally, research has characterized narcissism as a stable personality trait closely associated with an individual’s social behavior [6,7,8]. Unlike this perspective, the Dynamic Self-Regulatory Processing Model [9] reveals the situationally dependent nature of narcissism, referred to as state narcissism [10,11]. State narcissism refers to an individual’s temporary, heightened focus on and admiration of themselves in specific contexts [12,13], and this variability can significantly influence their behavioral expressions [14].

Existing research on the relationship between narcissism and prosocial behavior reveals a complex and varied landscape [8]. Some studies indicate no significant correlation between narcissism and prosocial behavior [15], while others suggest either a positive or negative correlation [1,16,17]. To enhance our understanding of this intricate relationship, the present study examines both trait and state narcissism, as well as the roles that situational and motivational factors play in their association with prosocial behavior, as outlined below.

First, this study examines the association between trait narcissism and prosocial behavior in various contexts, including both public and anonymous situations. Second, it explores the effect of state narcissism on prosocial behavior across different situations. Finally, this study investigates the mediating factor, specifically self-enhancement, that may influence the relationship between narcissism and prosocial behavior, thereby elucidating the potential mechanisms. By these studies, we aim to reveal how trait and state narcissism are related to behaviors sensitive to environmental cues, which contributes to a deeper understanding of the complex interplay between personality and environment in shaping social behavior.

### 1.1. Narcissism and Prosocial Behavior: The Impact of Situations

Prosocial behavior is typically defined as a category of positive social actions aimed at benefiting others, including activities such as donating, sharing, helping, volunteering, and giving [18]. A significant body of research addressing the relationship between narcissism and prosocial behavior specifically examines trait narcissism [19]. For instance, studies have shown that individuals with high levels of trait narcissism often demonstrate lower empathy and agreeableness [5,16], both of which are critical predictors of prosocial behavior. However, other research has identified instances where trait narcissism may positively correlate with prosocial behavior [17,20], indicating a more nuanced relationship than previously understood. The inconsistency of these findings may be attributed to the complex nature of narcissism and prosocial behaviors. For instance, individuals may engage in prosocial behavior for a variety of reasons, not through solely altruistic motives [1]. The psychological dynamics model of narcissism indicates that there are significant variations in narcissism across different backgrounds and situations [21], which may further influence subsequent behavioral outcomes.

Moreover, trait activation theory (TAT) posits that behavioral changes are, to some extent, influenced by cues associated with personality traits. However, it is crucial to acknowledge that personality traits, such as dutifulness, openness, and even narcissism, can be shaped by various situational factors operating on multiple levels [22]. In other words, these diverse situational characteristics also impact the expression of personality traits and their behavioral outcomes. For instance, research found that narcissists tend to demonstrate their prosocial behavior by posting videos of themselves participating in the Ice Bucket Challenge on public social networking sites, rather than opting to donate money to charity privately [1]. According to the TAT, public situational features can activate essential traits in narcissistic individuals, such as the excessive need for others to recognize their entitlement and superiority. When these traits are fully activated, they may lead to an increase in prosocial behaviors exhibited by narcissists. Similarly, another study indicated that narcissists self-reported a greater likelihood of engaging in prosocial behavior in the public workplace, compared to their less narcissistic colleagues [23].

In addition to trait narcissism, state narcissism also contributes to our understanding of the relationship between narcissism and prosocial behavior. To the best of our knowledge, there are only two studies currently available that address the relationship between state narcissism and behavior. One study suggests that self-reported state narcissism in daily life can positively predict helping behaviors [12], while another indicates that individuals with grandiose narcissism frequently communicate their various helping efforts, which are linked to an increase in their state of grandiose narcissism [24]. Given the correlation between trait and state narcissism [10,25], these constructs share similarities in predicting certain behavioral outcomes [10,11,12]. For instance, research has shown that consumers exhibiting high levels of trait or state narcissism tend to select products with more unique and distinctive features [14]. Therefore, it is reasonable to speculate that state and trait narcissism may similarly relate to prosocial behavior and that both are influenced by situational factors.

### 1.2. Self-Enhancement as a Potential Explanatory Mechanism

Building on the previous discussion, situational factors, including public and anonymous situations, can act as moderating variables that influence the relationship between narcissism and prosocial behavior. Additionally, motivational factors, particularly self-enhancement, may serve as an intrinsic mechanism through which narcissists engage in prosocial behavior.

Self-enhancement refers to the tendency to maintain unrealistically positive self-views [26,27]. According to the extended agency model of narcissism [28], the diverse self-regulatory strategies employed by narcissists—including admiration seeking, bragging, the display of material goals, and socializing with influential individuals—serve as both causes and consequences of their inflated self-beliefs. Narcissists heavily rely on self-enhancement strategies to bolster their egos [29]. Prior research indicates that narcissists prioritize self-enhancement more than others do [30]. Narcissism is likely associated with self-enhancement, as it embodies an extreme manifestation of the desire to “maintain, protect, and enhance one’s self-esteem” [31,32]. Furthermore, previous studies suggest that self-enhancement arises from specific experiences unique to narcissists [33,34]. Consequently, inducing state narcissism is likely to enhance an individual’s self-enhancement.

In addition, the self-enhancement of altruistic traits mediates the influence of interdependence on helping behaviors [35]. This suggests that self-enhancement is not solely about individual self-perception; it also affects one’s willingness to assist others in social interactions. Consequently, it is reasonable to speculate that when inducing a state of narcissism, individuals may display prosocial behavior as a result of self-enhancement, i.e., a positive self-perception [23]. However, it is essential to recognize that the impact of this relationship may vary across different situations [36]. As the behavioral outcomes of self-enhancement need others’ regard [26,27], public situations providing potential for reputation enhancement and social approval can intensify the effect of self-enhancement on prosocial behavior. Conversely, in anonymous situations without external recognition, even if the inclination towards self-enhancement is activated, individuals may opt for self-indulgent choices and thus display authentic behavior.

### 1.3. The Present Study

Based on the foregoing review, we hypothesize that public (vs. anonymous) situations would lead individuals with higher levels of trait narcissism or state narcissism to demonstrate more prosocial behaviors (Hypothesis 1), as well as that induced state narcissism will promote self-enhancement, which subsequently increases prosocial behaviors in public situations rather than in anonymous situations (Hypothesis 2). To test our hypotheses systematically, we conducted three studies, including a survey and experiments.

This study has achieved several expansions in the relationship between narcissism and prosocial behavior. Firstly, while previous research has explored the connection between trait and state narcissism and prosocial behavior, state narcissism has typically been measured through self-reports in everyday contexts [12]. In contrast, this study employs a laboratory-induced method to activate participants’ state narcissism, thereby providing methodological support for future research. Secondly, this study investigates the relationship between narcissism and prosocial behavior in both public and anonymous situations, offering empirical evidence to address previous inconsistencies [15,16] and enriching trait activation theory [22]. This exploration enhances our understanding of how situational factors moderate the relationship between personality traits or states and behavior. Finally, this study aims to validate the extended agency model of narcissism [28] by examining whether state narcissism impacts prosocial behavior through the strategy of self-enhancement. Through these contributions, this study seeks to provide new insights into the complex relationship between narcissism and prosocial behavior.

## 2. Study 1

In Study 1, we examined the correlation between trait narcissism and prosocial tendencies among college students. Utilizing a questionnaire survey methodology, we measured both trait narcissism and prosocial tendencies through self-reported scales. The measures of prosocial tendencies encompassed two dimensions: public prosocial tendency and anonymous prosocial tendency. Specifically, we predicted that individuals with higher levels of trait narcissism would demonstrate stronger public prosocial tendencies compared to anonymous prosocial tendencies.

### 2.1. Method

#### Participants and Procedure

In the absence of predetermined expectations regarding effect sizes, we aimed to recruit as many participants as possible. We utilized a convenience sampling method, resulting in the collection of 957 questionnaires from a mid-sized public university in China. Of these, 25 responses were excluded due to incompleteness, as they contained missing information exceeding 20%. Consequently, our final sample comprised 932 undergraduates (*M*_age_ = 20.11 ± 1.53; 705 males). A sensitivity analysis conducted using G*Power 3.1 indicated that the sample size can achieve an 80% probability of detecting effects with a size of *f*^2^ = 0.008 or greater, indicating that our sample was adequately powered to identify small effect sizes.

All participants provided written consent before the study. The survey was conducted in the classroom with an assistant. All studies were approved by the Committee of Protection of Subjects at the first author’s university. The procedures employed in all studies were conducted in accordance with the principles outlined in the Declaration of Helsinki.

### 2.2. Materials

#### 2.2.1. Trait Narcissism

Participants in this study completed the Narcissistic Personality Questionnaire (NPQ) [37], which comprises 20 items, such as “I like to be the center of attention”, α = 0.83. Each item is rated on a 7-point Likert scale, ranging from 1 (strongly disagree) to 7 (strongly agree). The scores for all items were averaged to derive an indicator of trait narcissism. Confirmatory factor analyses of this scale demonstrated a good fit, *χ*^2^/*df* = 2.79, CFI = 0.939, TLI = 0.913, RMSEA = 0.044, 90%CI [0.039, 0.049].

#### 2.2.2. Prosocial Tendencies

Public prosocial tendency was measured with four items from a subscale of the revised version of the Prosocial Tendencies Measure [38], for example, “I can help others best when people are watching me”, α = 0.82. Anonymous prosocial tendency was evaluated through five items [38], such as, “I think that helping others without them knowing is the best type of situation”, α = 0.80. All items were rated on a 5-point Likert scale ranging from 1 (does not describe me at all) to 5 (describes me greatly). In this study, the validity of the two subscales was found to be acceptable, with the following fit indices: for the public prosocial tendency subscale, *χ*^2^/*df* = 4.13, CFI = 0.991, TLI = 0.974, RMSEA = 0.058, 90%CI [0.021, 0.101]; and for the anonymous prosocial tendency subscale, *χ*^2^/*df* = 6.93, CFI = 0.967, TLI = 0.934, RMSEA = 0.080, 90%CI [0.056, 0.106].

### 2.3. Results and Discussion

In this study, the proportion of missing values did not exceed 5% of the total dataset. Given that all the variables followed a normal distribution, we chose to impute the missing data using their respective mean values. Furthermore, the Harman single-factor test was conducted to assess the presence of common method biases. A serious problem of common method bias is indicated if the first factor in factor analysis accounts for more than 40% of the variance in the covariance of the measurements. The results of this study reveal that the first factor accounted for only 17.09% of the variance, indicating no significant issue with common method bias.

The descriptive statistics and zero-order correlations among the study variables are shown in Table 1. A positive correlation was observed between trait narcissism and public prosocial tendency (*r* = 0.34, *p* < 0.001), while the correlation between trait narcissism and anonymous prosocial tendency was not significant (*r* = 0.03, *p* = 0.35).

Regressing prosocial tendency in the public and anonymous situations on trait narcissism yielded *Y_public_* = 1.81 + 0.45 *X* and *Y_anonymous_* = 3.41 + 0.04 *X*. In the first equation, trait narcissism was a significant predictor of prosocial tendency in the public situation, *t* = 10.92, *p* < 0.001, *R*^2^ = 0.114; in the second equation, the slope for trait narcissism was not significantly different from zero, *t* = 0.94, *p* = 0.35, *R*^2^ = 0.001. When the situation difference [39,40], *Y_D_* = *Y_anonymous_* − *Y_public_*, was regressed on trait narcissism, the following equation was obtained: *Y_D_* = 1.60 − 0.41 *X*. The slope for trait narcissism was significantly different from zero, *t* = −7.66, *p* < 0.001, *R*^2^ = 0.059. Thus, the results showed that the slope for trait narcissism in the public situation differed from the slope for trait narcissism in the anonymous situation. [Note: *Y_public_* = public prosocial tendency, *Y_anonymous_* = anonymous prosocial tendency, *X* = trait narcissism. The results of including age and gender as covariates were found to be similar to not controlling for them, see Appendix B].

Study 1 provides preliminary correlational evidence supporting our hypothesis that trait narcissism is related to prosocial tendencies in public but not anonymous situations. Given that narcissism encompasses both a process and a state component [10], Studies 2 and 3 employed experimental designs to examine the influence of state narcissism on prosocial behaviors across various situations.

## 3. Study 2

The primary objective of Study 2 was to explore the relationship between state narcissism and prosocial behaviors in public and anonymous situations. By utilizing an experimental approach, we aimed to provide stronger evidence for the relationship between narcissism and prosocial behaviors. We experimentally induced narcissism and control states (neutral affect) by using guided imagination manipulation—a well-validated technique for inducing narcissism states [13]. Participants’ prosocial behavior was measured by an actual monetary donation paradigm and was set in either a public or anonymous situation.

### 3.1. Method

#### Participants and Design

We performed a sample size calculation on the basis of G*Power 3.1 (linear multiple regression, fixed model, *R*^2^ increase) using an alpha level of 0.05 with 80% power to detect an effect size (*f*^2^ = 0.05). The results showed that a sample size of 159 was needed to ensure adequate statistical power. We recruited 200 undergraduates (93 males) from a mid-sized public university in western China to participate in our study. Participants were randomly assigned to one of four conditions in a 2 (group: state narcissism vs. neutral) × 2 (situation: public vs. anonymous) between-subjects factorial design.

### 3.2. Procedures and Materials

Upon arrival at the laboratory, participants were first presented with an informed consent form. Subsequently, they completed questionnaires designed to assess trait narcissism and prosocial behavior.

#### 3.2.1. Trait Narcissism

The Narcissistic Personality Inventory (NPI) [41] was used to assess participants’ trait narcissism with 40 items (e.g., “I am more competent than others”). The item scores were assessed on a 6-point Likert scale ranging from 1 (strongly disagree) to 6 (strongly agree), α = 0.91. Confirmatory factor analyses of this scale yielded a good fit, *χ*^2^/*df* = 1.26, CFI = 0.952, TLI = 0.935, RMSEA = 0.036, 90%CI [0.027, 0.044].

#### 3.2.2. Prosocial Tendencies

All participants completed the Prosocial Tendencies Measure [38]. In this study, the scores of the 26 items were averaged, and higher scores suggested greater prosocial tendency (α = 0.89). Each item was rated on a 5-point Likert scale, where 1 represented “does not describe me at all” and 5 indicated “describes me greatly”. The index of confirmatory factor analyses suggested strong structural validity: *χ*^2^/*df* = 1.17, CFI = 0.979, TLI = 0.971, RMSEA = 0.029, 90%CI [0.007, 0.043].

#### 3.2.3. Manipulation of the State Narcissism

One week later, each participant returned to the laboratory individually to engage in an imagination and relaxation task. Participants were randomly assigned by the server to either the narcissism manipulation group (*n* = 98) or the neutral condition group (*n* = 102). In the neutral group, male participants listened to the story of the protagonist Prometheus, while female participants were presented with the story of Aphrodite. Conversely, in the state narcissism group, male participants heard the story of a handsome prince, and female participants listened to the tale of a beautiful princess (see Appendix A) [11]. Participants were instructed to sit comfortably on a couch, close their eyes, and breathe deeply. The duration of each story was approximately three minutes, during which participants were encouraged to vividly imagine the scenes depicted.

#### 3.2.4. Manipulation Check

We adapted the Narcissistic Admiration and Rivalry Questionnaire (NARQ) to assess the state narcissism [42]. Participants rated their agreement (1 = strongly disagree; 6 = strongly agree) with 18 items measuring two subdimensions: narcissistic admiration (e.g., “I am great”) and narcissistic rivalry (e.g., “I react annoyed if another person steals the show from me”). The Cronbach’s alpha values were 0.85 for the narcissistic admiration dimension and 0.81 for the narcissistic rivalry dimension. This scale demonstrated good validity in the current study, *χ*^2^/*df* = 1.77, CFI = 0.924, TLI = 0.903, RMSEA = 0.062, 90%CI [0.048, 0.070].

#### 3.2.5. Prosocial Behavior (Monetary Donations)

After the task concluded, participants were given an envelope containing ten pieces of RMB 1 as compensation for their involvement in the study. This measure was implemented to prevent participants from associating the donation with the above task and to ensure they had funds available for donation. Subsequently, another experimenter, who was a volunteer of a charity and unaware of the design and purpose of the experiment, knocked on the door, entered the room, and introduced that her charitable organization was raising money for homeless people. At this stage, the participants were assigned to one of two conditions, where they were offered the opportunity to donate their compensation to a nonprofit organization either publicly (*n* = 100) or anonymously (*n* = 100). In the public situation, the confederate walked into the laboratory with a box; participants who wanted to donate could directly insert the donated money into the box. In the anonymous situation, the confederate came into the room without a box; participants could keep the money they wanted to donate in the envelope individually and anonymously in a nearby room. The monetary contributions made by participants served as a measure of prosocial behavior.

Following the final donation allocation session, participants were queried about whether they had discerned the purpose of the study. No participant guessed the true purpose of the current study. Finally, participants were debriefed, thanked, and asked to keep the procedure confidential.

### 3.3. Results and Discussion

#### 3.3.1. Manipulation Check Results

A *t* test indicated that participants in the state narcissism group (*M* = 4.01, *SD* = 0.63) reported more narcissistic admiration than those in the neutral group (*M* = 3.74, *SD* = 0.70), *t* (198) = 2.85, *p* = 0.005, 95%CI (confidence interval) = [0.08, 0.46], and there was no significant difference in reported narcissistic rivalry between the state narcissism group (*M* = 3.14, *SD* = 0.69) and the neutral group (*M* = 3.05, *SD* = 0.74), *t* (198) = 0.90, *p* = 0.37, 95%CI = [−0.10, 0.29], providing evidence for the effectiveness of our manipulation on state narcissism.

#### 3.3.2. Correlational Analysis

Table 2 shows the means, standard deviations, and correlations for trait narcissism, prosocial tendency, gender (male = 1, female = 0), group (state narcissism group = 1, neutral group = 0), situation (public = 1, anonymous = 0), and prosocial behavior (monetary donations) in Study 2.

#### 3.3.3. Main Analyses

Regression analyses were conducted with and without control variables (such as trait narcissism, prosocial tendencies, and gender), and the results were nearly identical. For simplicity, we did not include control variables in the main analyses in the manuscript. However, the results of the regression analyses with the addition of control variables are provided in Appendix B.

The interaction between group and situation was significant, *B* = 2.04, *SE* = 0.60, 95%CI = [0.86, 3.21], *t*(196) = 3.43, *p* < 0.001, *F*(3, 196) = 33.97, *R*^2^ = 0.34. To test our prediction that state-narcissistic people exhibit more prosocial behaviors in public situations, the situation was considered as the moderator, and simple slope analyses were used to test the effect of group (predictor) on prosocial behaviors (dependent variable) for each situation (moderator). As shown in Figure 1, the simple slope analyses indicated that state narcissism predicted more donations in the public situation, *B* = 2.42, *SE* = 0.42, 95%CI = [1.59, 3.25], *t*(196) = 5.76, *p* < 0.001, but there was a nonsignificant trend in the anonymous situation, *B* = 0.38, *SE* = 0.42, 95%CI = [−0.45, 1.21], *t*(196) = 0.91, *p* = 0.36.

The results also showed that there was a significant main effect of group, *B* = 1.40, *SE* = 0.30, 95%CI = [0.82, 1.99], *t*(196) = 4.71, *p* < 0.001, indicating that participants in the state narcissism group were more prosocial than participants in the neutral condition. There was also a significant main effect of situation, *B* = 2.42, *SE* = 0.30, 95%CI = [1.84, 3.01], *t*(196) = 8.12, *p* < 0.001, in which the participants in the public situation made greater donations than those in the anonymous situation.

The results of Study 2 indicated that individuals with state narcissism exhibit more prosocial behaviors in public situations compared to anonymous situations. This finding provides further evidence that situational factors significantly influence the relationship between narcissism and prosocial behavior. However, an important question remains: Why do narcissistic individuals engage in more prosocial behavior in public situations? There are several potential explanations for this phenomenon. According to the extended agency model [28], the narcissistic tendency for self-enhancement may drive their increased prosocial behavior, a process that is also shaped by situational influences. In public situations, the desire for social approval and the potential for reputation enhancement can amplify the impact of self-enhancement on prosocial actions. Furthermore, Study 3 developed a moderated mediation model (Figure 2) to explore the relationship between narcissism and prosocial behavior in greater depth.

## 4. Study 3

This study aimed to investigate the mechanism through which state narcissism influences prosocial behavior. We hypothesized that experiences of state narcissism enhance prosocial behavior via self-enhancement and that this process is moderated by the situation.

### 4.1. Method

#### Participants and Design

We determined the prior sample size required for an effect size of interest (*f*^2^ = 0.05) by G*power 3.1 (linear multiple regression, fixed model, *R*^2^ increase); at least 159 participants were required to achieve 80% power (α = 0.05). In this study, 170 undergraduates (*M_age_* = 18.42 ± 1.08; 78 males) from a mid-sized public university in western China completed the experiment in a 2 (group: narcissism vs. neutral) × 2 (situation: public vs. anonymous) between-participant factorial design.

### 4.2. Procedures and Materials

In light of the observed similarities between the results of utilizing trait narcissism and prosocial tendencies as control variables and the outcomes when these variables were not employed in Study 2, Study 3 did not measure participants’ trait narcissism and prosocial tendencies. Participants individually arrived at the laboratory and provided consent before completing basic demographics.

#### 4.2.1. Manipulation of State Narcissism

Consistent with Study 2, participants were randomly assigned to either the narcissism manipulation group (*n* = 86) or the neutral condition group (*n* = 84). Specifically, the participants were informed that they were going to listen to a brief story (imagination and relaxation task). Subsequently, they were asked to respond to a series of questions aimed at assessing their state narcissism and self-enhancement. To conclude, participants were asked to participate in a donation experiment identical to that of Study 2 in both public (*n* = 81) and anonymous (*n* = 89) situations.

#### 4.2.2. Manipulation Check

We adapted the Narcissistic Admiration and Rivalry Questionnaire (NARQ) to assess state narcissism [42], which is consistent with the scale used in Study 2. The scores on this scale yielded an alpha of 0.79 for narcissistic admiration and 0.67 for narcissistic rivalry in this study. Furthermore, confirmatory factor analyses were employed in this study, and the results suggested an acceptable structural validity: *χ*^2^/*df* = 1.47, CFI = 0.918, TLI = 0.900, RMSEA = 0.053, 90%CI [0.036, 0.069].

#### 4.2.3. Self-Enhancement

To assess self-enhancement, we used 32-adjective phrases referring to self-enhancement [43,44]. The self-enhancement dimension included “confidence” and “friendly”, and participants then rated how confident or friendly they were from 1 (“I am less confident/friendly than others”) to 7 (“I am more confident/friendly than others”). The Cronbach’s alpha for this scale was 0.87. Confirmatory factor analyses of this scale yielded an acceptable fit, *χ*^2^/*df* = 1.50, CFI = 0.930, TLI = 0.904, RMSEA = 0.055, 90%CI [0.045, 0.064].

#### 4.2.4. Prosocial Behavior (Monetary Donations)

The measurement of prosocial behavior in this study is consistent with that in Study 2.

### 4.3. Results and Discussion

#### 4.3.1. Manipulation Check

A *t* test indicated that participants in the state narcissism condition (*M* = 3.99, *SD* = 0.68) reported more narcissistic admiration than those in the neutral condition (*M* = 3.49, *SD* = 0.49), *t* (168) = 5.51, *p* < 0.001, 95%CI = [0.32, 0.68], and there was no significant difference in reported narcissistic rivalry between the state narcissism group (*M* = 2.98, *SD* = 0.83) and neutral group (*M* = 2.80, *SD* = 0.60), *t* (168) = 1.57, *p* = 0.12, 95%CI = [−0.05, 0.39], providing evidence for the effectiveness of our manipulation on state narcissism.

#### 4.3.2. Correlational Analysis

Table 3 shows the means, standard deviations, and correlations for gender, age, group, situation, self-enhancement, and prosocial behavior (monetary donations) in Study 3.

#### 4.3.3. Main Analysis

We examined whether differences in feelings of self-enhancement could explain why state-narcissistic participants in the anonymous situation did not experience the same boost in prosocial behaviors as narcissistic participants in the public condition. To achieve this, we used a bootstrap analysis with 5000 samples for moderated mediation, treating self-enhancement as a mediator and situation (public, anonymous) as a moderator (Process Model 14); please refer to Table 4 and Figure 3 for details.

The results indicated that the interaction between self-enhancement and situation significantly predicted prosocial behavior (*B* = 2.21, *SE* = 0.64, 95% CI = [1.12, 3.23], *t*(166) = 3.45, *p* = 0.001), suggesting that narcissism influences prosocial behavior through self-enhancement, moderated by situation. To reveal how the situation moderates the impact of self-enhancement on prosocial behavior, we conducted a simple slope analysis. As shown in Figure 3, the simple slope analyses indicated that self-enhancement predicted more donations in the public situation (*B* = 1.55, *SE* = 0.46, 95% CI = [0.64, 2.47], *t*(166) = 3.35, *p* < 0.001), but there was a nonsignificant trend in the anonymous situation (*B* = −0.70, *SE* = 0.53, 95%CI = [−1.75, 0.40], *t*(166) = −1.30, *p* = 0.20).

This pattern supports our hypothesis that state narcissism increases self-enhancement, which mediates the positive effect of state narcissism on monetary donations. However, it is important to acknowledge that the role of the situation cannot be ignored in this process. In public situations, state narcissism promotes more prosocial behavior through self-enhancement, while in anonymous situations, this trend is not significant. This also reflects that narcissists’ self-enhancement may also be a kind of self-presentation, and their starting point for prosocial behaviors is still for their own sake.

## 5. General Discussion

The present research investigated the relationship between narcissism and prosocial behaviors, focusing on the conditional influence of (public versus anonymous) situations and the potential mediating role of self-enhancement. The findings from all three studies indicated that individuals with higher levels of narcissism—both trait and state—tend to engage in more prosocial behaviors in public situations compared to anonymous situations. Furthermore, the results highlighted the mediating effect of self-enhancement on the relationship between state narcissism and prosocial behavior.

### 5.1. Theoretical Implications

This research contributes to our understanding of the relationship between narcissism and prosocial behaviors. The research revealed that in public situations, individuals with trait or state narcissism are likely to leverage these environments to showcase their prosocial behaviors. This finding extends beyond prior studies that primarily examined the relationship between stable trait narcissism and subjective prosocial behavior through self-report measures [12]. Moreover, by effectively stimulating state narcissism, this study clearly illustrated that in specific contexts, this immediate surge in narcissistic sentiment can positively motivate individuals to engage in prosocial behaviors. This indicates that the relationship between narcissism and objective prosocial behavior can be significantly shaped by public circumstances. This discovery offers a fresh perspective on the inconsistencies observed in the relationship between narcissism and prosocial behavior in earlier research, suggesting that while a connection exists, it is not unconditional and is profoundly influenced by situational factors.

This research also enhanced our understanding of the relationship by uncovering more nuanced underlying mechanisms. This study innovatively elucidates the critical role of self-enhancement as a mediator between state narcissism and prosocial donation behavior, while also uncovering the moderating influence of situations on this relationship. These findings challenge the conventional belief that prosocial behavior is solely driven by altruism [45], indicating that such behavior may not always originate from selfless intentions. In particular, for individuals exhibiting narcissistic traits, their prosocial actions may be more closely linked to self-enhancement and contextual influences rather than genuine altruism. This mechanism offers a valuable framework for understanding how narcissistic personalities can foster prosocial behaviors and validates the extended agency model of narcissism [28]. Although narcissists engage in prosocial activities, their motivations may be rooted in self-serving interests [46], such as self-enhancement, particularly in public situations.

In summary, the findings of this research indicate that narcissists’ prosocial behaviors in public situations demonstrate their adeptness in utilizing various strategies to achieve or maintain their grandiose self-esteem [9]. According to the extended agency model [28], narcissism can be regarded as a prioritization of agency—such as status and power—over communion. This prioritization leads to distinct interpersonal strategies, such as self-enhancement, and intrapersonal strategies, including self-serving biases. Narcissists, who emphasize agentic concerns, are likely to engage in prosocial behaviors only when they perceive a potential benefit, whether through direct reciprocation or intangible rewards. Our research findings bolster this perspective, indicating that the interplay among trait narcissism, state narcissism, and prosocial behavior is significantly influenced by situational factors, particularly as participants recognize that exhibiting prosocial behavior in public contexts may garner praise from others and serve their self-interests.

Furthermore, the present research furthers our understanding of narcissism. On the one hand, our findings demonstrate that both trait and state narcissism exhibit similar patterns in their relationship with prosocial behavior. Specifically, both forms of narcissism reflect underlying tendencies that operate at different levels and collectively shape an individual’s behavior in particular contexts. This also confirms the perspective of Trait Activation Theory, which posits that narcissism may be affected by various situational factors operating at multiple levels [22]. While trait narcissism may serve as a predictor for general behavioral trends, state narcissism appears to exert a more immediate impact on individuals’ behavioral responses. Therefore, to effectively understand and predict the prosocial behavior of narcissistic individuals, it is essential to consider not only their stable levels of trait narcissism but also the ways in which the current situation may activate or inhibit their state narcissism. This comprehensive approach is necessary to unravel the intricate relationship between narcissism and prosocial behavior.

On the other hand, our findings indicate that the activation of state narcissism implies that stable personality traits can be influenced by situational factors and may undergo changes. This finding suggests that an individual’s behavioral performance and psychological state are not solely determined by their inherent personality traits; rather, situational factors also play a crucial role in either stimulating or inhibiting the expression of specific traits. This perspective offers new insights into the relationship between personality traits and behavior.

### 5.2. Practical Implications

This research also has practical implications by providing effective strategies that encourage narcissists to engage in prosocial behaviors. One promising method is ‘situational design’, which recognizes that narcissists are more likely to exhibit prosocial behaviors in public situations. By organizing public charity activities or challenges, we can create compelling incentives for their participation, thereby fostering their prosocial actions. Additionally, mobilizing social media can serve as an effective strategy, as narcissists often seek attention and admiration on these platforms [3]. By launching prosocial campaigns, we can pique their interest and promote their active involvement. Furthermore, since narcissists typically engage in self-construction and self-enhancement through social relationships [9,47], organizing charitable activities that allow them to take on leadership roles or provide opportunities for public speaking can effectively fulfill their need for attention and admiration. This, in turn, can lead to a sense of self-enhancement and encourage further prosocial behaviors. By strategically employing these approaches, we can harness the tendencies of narcissists and channel them towards making positive social contributions.

### 5.3. Strengths and Limitations

The present research contributes to the literature in several ways. First, it addresses previous inconsistencies in conclusions regarding the relationship between narcissism and prosocial behavior. Given the complexities inherent in both constructs, a thorough understanding of their connection necessitates consideration of situational factors. The relationship between narcissism and prosocial behavior may manifest differently across various situations. Second, our study adopted a more comprehensive approach to measure prosocial behaviors compared to prior research, incorporating self-reports of prosocial tendencies in diverse situations alongside monetary donations to charitable organizations. Third, we also examined the psychological reasons for narcissists’ prosocial behaviors, that is, the situations and motives that elicit these behaviors. Most importantly, we examined the relationship between state narcissism and prosocial behavior and found that it is similar to the relationship between trait narcissism and prosocial behavior. This result also provides promising evidence for the conceptualization of narcissism on a state level [25]. We believe that understanding the complex dynamic system of narcissism requires studying it from moment to moment and across situations [48]. Our studies provide robust empirical evidence for the notion that heightened state narcissism contributes to increased prosocial behaviors in public, with self-enhancement playing a crucial role in this mechanism.

However, there are also limitations and additional questions that need to be addressed in future research. The primary limitation of our study lies in the exclusive recruitment of undergraduate students from west China, which restricts the demographic and developmental diversity of the sample. Future studies should aim to include a more varied participant pool to enhance the generalizability of the findings. Second, the current study did not measure the duration of state narcissism induced by our primes, and future research could focus on the intensity and duration of state narcissism. Moreover, this study did not examine whether the observed effects of narcissism on prosocial behavior were stable over time, highlighting the need for longitudinal research in subsequent studies. Finally, it is essential for future research to incorporate appropriate attention-checking questions after participants engage with the story, such as inquiring about the name of the main character, to more rigorously assess participants’ attention levels.

## 6. Conclusions

This study examines the relationship between narcissism and prosocial behavior, emphasizing the mediating role of self-enhancement and the moderating effects of public versus anonymous situations. The findings reveal that both trait and state narcissism are positively correlated with increased prosocial behaviors. Additionally, the application of self-enhancement strategies plays a crucial role in the mechanism through which state narcissism affects prosocial behavior. Notably, these effects are significantly pronounced only in public situations.

## Figures and Tables

**Figure 1 behavsci-14-01200-f001:**
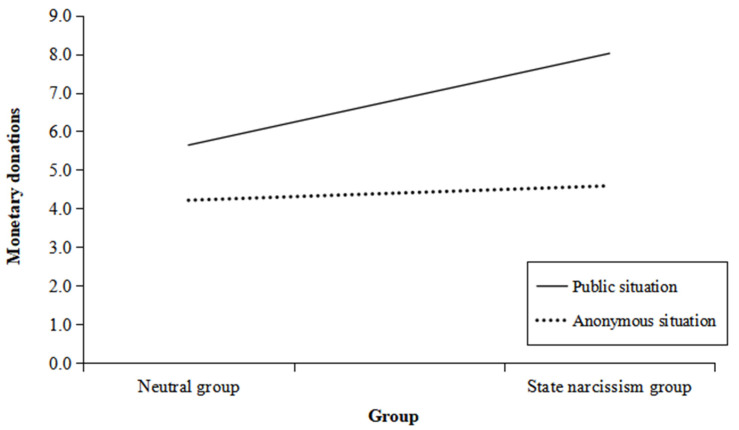
Prosocial behaviors (monetary donations) as a function of situations and groups of Study 2.

**Figure 2 behavsci-14-01200-f002:**
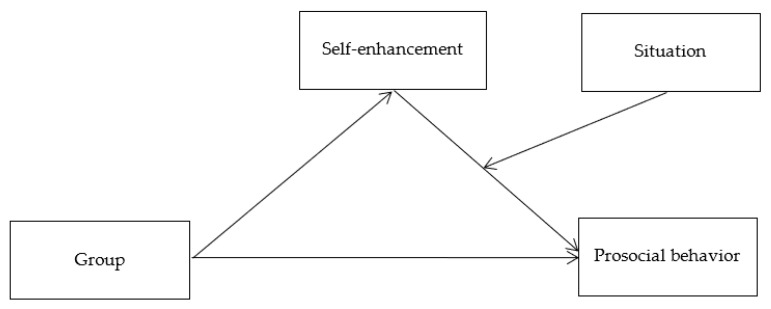
A moderated mediation model, in which narcissism primes (group) are hypothesized to influence donations (prosocial behavior) via self-enhancement.

**Figure 3 behavsci-14-01200-f003:**
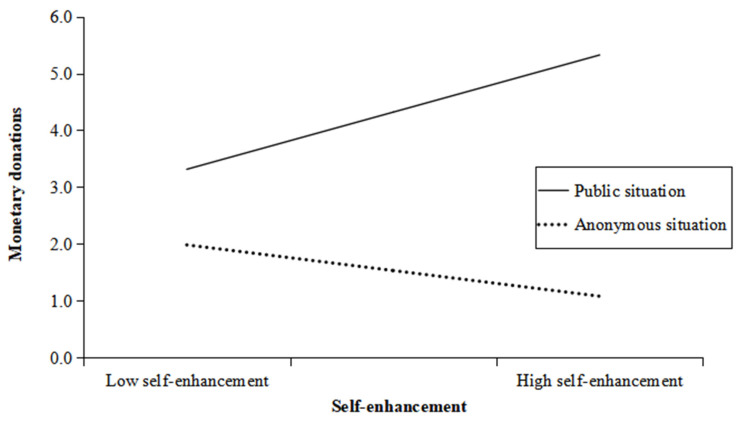
Prosocial behaviors (monetary donations) as a function of situations and self-enhancement of Study 3.

**Table 1 behavsci-14-01200-t001:** Means, standard deviations, and correlations among the study variables (Study 1, *n* = 932).

Variables	*M*	*SD*	1	2	3	4	5
1. Trait narcissism	3.03	0.50	1				
2. Public prosocial tendency	3.18	0.67	0.34 ***	1			
3. Anonymous prosocial tendency	3.53	0.61	0.03	0.12 ***	1		
4. Gender	-	-	0.05	0.04	−0.07 *	1	
5. Age	20.11	1.53	−0.07 *	−0.08 *	−0.04	0.04	1

Note. *** *p* < 0.001, * *p* < 0.05. Gender coded (males = 1, females = 0).

**Table 2 behavsci-14-01200-t002:** Means, standard deviations, and correlations among the study variables (Study 2, *n* = 200).

Variables	*M*	*SD*	1	2	3	4	5	6
1. Trait narcissism	3.53	0.53	1					
2. Prosocial tendency	3.69	0.44	0.24 **	1				
3. Gender	-	-	0.16 *	−0.13	1			
4. Group	-	-	−0.11	−0.04	−0.05	1		
5. Situation	-	-	−0.17 *	0.02	0.05	0.02	1	
6. Monetary donations	3.51	2.57	−0.14	0.06	0.13	0.28 ***	0.48 ***	1

Note. *** *p* < 0.001, ** *p* < 0.01, * *p* < 0.05. Gender coded (males = 1, females = 0), group coded (state narcissism group = 1, neutral group = 0), situation coded (public = 1, anonymous = 0).

**Table 3 behavsci-14-01200-t003:** Means, standard deviations, and correlations among the study variables (Study 3, *n* = 170).

Variables	*M*	*SD*	1	2	3	4	5	6
1. Gender	-	-	1					
2. Age	18.42	1.08	0.20 **	1				
3. Group	-	-	−0.06	0.04	1			
4. Situation	-	-	−0.004	0.12	0.02	1		
5. Self-enhancement	5.02	0.65	−0.13	0.15	0.49 ***	0.22 **	1	
6. Monetary donations	3.08	3.31	−0.14	−0.03	0.15	0.44 ***	0.21 **	1

Note. *** *p* < 0.001, ** *p* < 0.01. Gender coded (males = 1, females = 0), group coded (state narcissism group = 1, neutral group = 0), situation coded (public = 1, anonymous = 0).

**Table 4 behavsci-14-01200-t004:** The results of moderated mediation analyses in Study 3.

Outcomes		Effect	*SE*	*t*	95%CI	*F*	*R* ^2^
Self-enhancement	Predicting variable						
	Group	0.63	0.09	7.34 ***	[0.49, 0.77]	53.10	0.24
Donations	Predicting variables						
	Group	0.64	0.50	1.28	[−0.17, 1.46]	14.68	0.32
	Self-enhancement	−0.89	0.40	−2.21 *	[−1.58, −0.25]		
	Situation	−8.28	3.25	−2.55 *	[−13.55, −2.88]		
	Self-enhancement × Situation	2.21	0.64	3.45 **	[1.12, 3.23]		

Note. * *p* < 0.05, ** *p* < 0.01, *** *p* < 0.001. Group coded (state narcissism group = 1, neutral group = 0), situation coded (public = 1, anonymous = 0).

## Data Availability

Due to confidentiality/privacy concerns, the data used in this study are not publicly available at the moment. Interested researchers can contact the authors for further information or access to the data.

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
