# Peer review of "I’m More Prosocial than Others: Narcissism Facilitates Prosocial Behavior in Public Situations"

_behavsci, 2024, doi:10.3390/bs14121200_

Round 1

Reviewer 1 Report

Comments and Suggestions for Authors

Thank you very much for the opportunity to revise this manuscript covering the association between narcissism and pro-social practices. The study relies on three studies in which narcissism plays a role in pro-social behaviours in public vs anonymous situations. Although the work addresses an interesting topic, I have some concerns about this manuscript. Below are my comments point-by-point, which I hope can enhance the overall quality of this paper.

Abstract I suggest authors explain more about the three studies from a methodological point of view (e.g.,  the N for each study). Also, results should be disclosed better for each study.

Introduction Lines 23-33. The authors described the phenomenon of “vanity charity” that, using the authors’ words, involves “sharing information about their charitable acts, not out of genuine care and concern for others but rather to seek praise and enhance their own image”. I understand the logic of this phenomenon, but the authors did not provide a clear scientific perspective about vanity charity. Also, no citations about this phenomenon have been provided. Then, I believe that some bold claims should be rephrased. For instance, “This phenomenon of “vanity charity” highlights the complex interplay between self-presentation and altruistic behaviour, which also shows a narcissistic mentality to a certain extent”. I do not understand the meaning of narcissistic mentality here. Please explain better and use the terms appropriately. Then, in this pre-intro lines 34-56, I suggest improving the description of the main features trait narcissism and state narcissism, highlighting the differences between them. Maybe the information provided in the 1.4 could be included here, dropping the 1.4. I suggest some literature that needs to be included and described: De Bellis, E., Sprott, D. E., Herrmann, A., Bierhoff, H. W., & Rohmann, E. (2016). The influence of trait and state narcissism on the uniqueness of mass-customized products. Journal of Retailing, 92(2), 162-172. Giacomin, M., & Jordan, C. H. (2018). State narcissism. Handbook of trait narcissism: Key advances, research methods, and controversies, 105-111. Giancola, M., Perazzini, M., Bontempo, D., Perilli, E., & D’Amico, S. (2024). Narcissism and Problematic Social Media Use: A Moderated Mediation Analysis of Fear of Missing Out and Trait Mindfulness in Youth. International Journal of Human–Computer Interaction, 1-11. Then, at the end of this pre-into (lines 34-56), the author should describe better their aims. I found the paragraph division to be very helpful. However, they have to guide the reader to understand the logic behind the article. In this case, I have some issues. For instance, the authors described the association between narcissism and pro-social behaviours. Then, we have the second paragraph titled “the impact of situations”. At first sight, I do not understand in which areas situations have a role. Please change the title of the paragraph or include 1.2 in 1.1. Then, as the authors use state and trait narcissism, I expect that the literature review on narcissism and pro-social behavious relies on both facets of narcissism. If possible, please clarify this distinction in all paragraphs of the introduction and throughout the entire manuscript (when necessary).   The mediating role of self-enhancement seems to go out of nowhere. Why did the author focus on this variable? They mentioned this concept only in the abstract. Maybe a brief definition of the pre-intro could be useful to improve the readability of the manuscript.    As for “the present study section”, the authors should better describe the logic behind their hypothesis from a theoretical point of view, explaining WHY they expect these relationships and effects. Then, the information provided on lines 180-189 are not necessary here! Also, the authors should express better the novelty of their work.

Method The authors should provide more information about data gathering for each study. Were there any attention checks in the survey? Were there any missing data? Can the authors provide more information about the drop-out rate? I suggest reporting in detail the procedure for data gathering. Please report the psychometric properties, features, and information for each questionnaire used in this research.

Results By reading the introduction, I understood that the authors also advanced a mediation model. However, by reading the results section I noted that the authors also provided a moderated mediation model which sees to go out of nowhere. The logic is OK, but the authors should explain this model from the introduction.

Discussion I believe that the discussion is totally unbalanced. The authors should discuss in detail their results instead of focusing on the limitations.  

Comments on the Quality of English Language

Authors should review the English moderately.

Reviewer 2 Report

Comments and Suggestions for Authors

Thank you for the opportunity to review this manuscript. This study examines the relationship between narcissism and prosocial behavior, with a particular focus on the role of situational factors and the effect of self-enhancement. Below, I provide several suggestions for improvement of the study. I hope these comments are helpful, and I wish the authors the best of luck in refining their work.

Introduction

1. The introduction begins with an anecdotal example of "vanity charity" on social media, but this is not systematically linked to the empirical literature on narcissism or prosocial behavior. While engaging, this example lacks generalizability and might not adequately represent the broader scope of narcissistic prosocial behaviors.

2. While the introduction claims that narcissistic prosocial behavior has received "less attention," it does not convincingly justify why this gap is important or how addressing it would advance the field. I would try to articulate a clearer rationale for why understanding narcissistic prosocial behavior is meaningful. Discuss how examining state narcissism adds value, such as revealing transient, context-sensitive behaviors versus enduring personality traits.

Study 3

3. The Cronbach's alpha for narcissistic rivalry (α = 0.67) falls below the commonly accepted threshold of 0.70, raising concerns about the reliability of this measure. Consider conducting additional validation of the scale within the sample.

4. The manuscript provides results from bootstrap analyses but does not include details such as model fit indices. This lack of information makes it challenging to assess the robustness of the findings. If I recall correctly, it is not possible to obtain model fit indices using SPSS Process Model 14. Is there a specific reason for not using structural equation modeling (SEM) instead? SEM offers several advantages over SPSS Process Models, including the ability to provide model fit indices and more flexibility in testing complex models.

General Discussion

5. Additionally, the discussion section is relatively brief, limiting the depth of analysis and contextualization of the results within existing theories.

6. Similarly, while the discussion references some prior research (e.g., Konrath et al., 2016; Wetzel et al., 2017), I believe that it does not exhaustively integrate the findings with the broader theoretical landscape, such as addressing inconsistencies or gaps in the literature.

Theoretical and Practical Implications

7. The practical implications are presented in broad terms, such as "we should find a way to motivate narcissists to engage in more prosocial behaviors," without specifying actionable strategies. You should propose concrete interventions or mechanisms that could leverage narcissism for prosocial outcomes.

Strengths and Limitations

8. The exclusive use of undergraduate students of west China as participants severely limits the demographic and developmental diversity of the sample.

9. Furthermore, the study does not explore whether the observed effects of narcissism and prosocial behavior are stable over time, so longitudinal research might be suggested for future studies.

Round 2

Reviewer 1 Report

Comments and Suggestions for Authors

All comments have been addressed. The manuscript is suitable for publication.

Reviewer 2 Report

Comments and Suggestions for Authors

I believe the manuscript is now worthy of publication.